# Impact of Peripheral Hydrogen Bond on Electronic Properties of the Primary Acceptor Chlorophyll in the Reaction Center of Photosystem I

**DOI:** 10.3390/ijms25094815

**Published:** 2024-04-28

**Authors:** Lujun Luo, Antoine P. Martin, Elijah K. Tandoh, Andrei Chistoserdov, Lyudmila V. Slipchenko, Sergei Savikhin, Wu Xu

**Affiliations:** 1Department of Chemistry, University of Louisiana at Lafayette, Lafayette, LA 70504, USA; luolujun56@163.com (L.L.);; 2Department of Physics, Purdue University, West Lafayette, IN 47907, USA; 3Department of Biology, University of Louisiana at Lafayette, Lafayette, LA 70504, USA; 4Department of Chemistry, Purdue University, West Lafayette, IN 47907, USA; lslipchenko@purdue.edu

**Keywords:** photosystem I, electron transfer, hydrogen bond, site-directed mutagenesis, ultrafast spectroscopy, chlorophyll

## Abstract

Photosystem I (PS I) is a photosynthetic pigment–protein complex that absorbs light and uses the absorbed energy to initiate electron transfer. Electron transfer has been shown to occur concurrently along two (A- and B-) branches of reaction center (RC) cofactors. The electron transfer chain originates from a special pair of chlorophyll *a* molecules (P700), followed by two chlorophylls and one phylloquinone in each branch (denoted as A_−1_, A_0_, A_1_, respectively), converging in a single iron–sulfur complex F_x_. While there is a consensus that the ultimate electron donor–acceptor pair is P700^+^A_0_^−^, the involvement of A_−1_ in electron transfer, as well as the mechanism of the very first step in the charge separation sequence, has been under debate. To resolve this question, multiple groups have targeted electron transfer cofactors by site-directed mutations. In this work, the peripheral hydrogen bonds to keto groups of A_0_ chlorophylls have been disrupted by mutagenesis. Four mutants were generated: PsaA-Y692F; PsaB-Y667F; PsaB-Y667A; and a double mutant PsaA-Y692F/PsaB-Y667F. Contrary to expectations, but in agreement with density functional theory modeling, the removal of the hydrogen bond by Tyr → Phe substitution was found to have a negligible effect on redox potentials and optical absorption spectra of respective chlorophylls. In contrast, Tyr → Ala substitution was shown to have a fatal effect on the PS I function. It is thus inferred that PsaA-Y692 and PsaB-Y667 residues have primarily structural significance, and their ability to coordinate respective chlorophylls in electron transfer via hydrogen bond plays a minor role.

## 1. Introduction

In proteins containing organic and inorganic cofactors, the protein environment modulates the properties of the cofactor. This generalization also applies to photosynthetic reaction centers, which are rich in diverse embedded cofactors such as chlorophylls (Chls), carotenoids, quinones, and iron–sulfur clusters. Many of these otherwise chemically identical cofactors have distinct spectroscopic and functional properties when embedded in protein environments. For example, photosystem I (PS I), the light-driven ferredoxin-plastocyanin oxidoreductase of cyanobacteria, contains 96 Chl *a* molecules, many of which form pools with distinct arrangements and properties [1,2,3]. Six of these Chl *a* molecules form a reaction center (RC) that initiates electron transfer (ET) processes [2], whereas the rest function in harvesting light and transferring excitation energy to RC [4,5]. Most of these Chl *a* molecules are chemically indistinguishable, but their individual electronic properties are fine-tuned by the surrounding protein to optimize light-harvesting and consequent charge transfer functions of PS I. The structural determinants responsible for the distinct properties of these Chl *a* molecules are not completely understood.

An increasing number of PS I structures of cyanobacteria, green algae, and plants that use white or far-red light or are in the monomer, trimer, and tetramer forms have been solved [2,6,7,8,9,10,11,12,13,14,15,16,17,18,19,20,21,22,23,24,25,26,27], showing a high resemblance between structures. Upon absorption of light, electronic excitation is created on one of the ~100 Chl *a* molecules followed by a sequence of rapid energy transfer steps that brings excitation to RC within ~20–60 ps [28,29,30,31,32] and initiates charge separation. The RC consists of six Chl molecules, arranged in two branches with a pseudo-C_2_ symmetry: a closely spaced special pair of Chls (P700), two A_−1_, two A_0_ Chls followed by a pair of phylloquinones A_1_ and merging at the iron–sulfur complex F_x_ (Figure 1). There is a consensus that in PS I, ET occurs concurrently along both branches of RC with some variation in the ratio of ET between the branches depending on species, though in all reported cases, A-branch (i.e., A_−1A_A_0A_A_1A_) was found to be preferable for ET [33,34,35,36]. All current models agree that upon excitation of RC either directly or via antenna, the charge-separated state P700^+^A_0_^-^ is formed within 0.1–3 ps (intrinsic rate) [37]. The electron is then transferred to A_1_ of the respective branch within 10–50 ps [5,38,39], followed by a much slower ET to F_x_, with transfer times being ~200 ns and ~20 ns for branches A and B, respectively [33,34]. The electron from F_X_ proceeds thereafter to iron–sulfur complexes F_A_ and F_B_ in ~500 ns [38], after which it is donated to soluble ferredoxin or flavodoxin. Reduced ferredoxin in normal conditions or flavodoxin, when the iron is scarce in the medium, [40] transfers the electron to ferredoxin-NADP^+^ reductase [41], which generates NADPH. P700^+^, in turn, is reduced on a much slower timescale by plastocyanin or soluble cytochrome *c*_6_.

In spite of three decades of research, there is still no agreement on the sequence and exact mechanism of the initial ET step involving six RC Chl molecules leading to the formation of the P700^+^A_0_^−^ state. In the most widely mentioned model, ET is initiated from the excited special pair (P700*) and is sequentially transferred to the accessory pigment A_−1_ and, then, to A_0_, forming the P700^+^A_0_^−^ redox pair [37]. Alternatively, it is proposed that A_−1_ rather than P700 is the primary electron donor, with the first charge-separated state being A_−1_^+^ A_0_^−^ [42,43,44]. Finally, it has also been proposed that either four (P700 and A_−1_) or all six RC Chls act as one supermolecule with electronic excitation delocalized over all pigments, initiating charge separation across that supermolecule and leading to the P700^+^A_0_^−^ state with no clearly identifiable intermediates [45,46,47]. Since primary charge separation in RC is much faster than the excitation transfer processes within the antenna and to RC, and since the absorption of RC cofactors overlaps with the absorption of numerous antenna Chls, the direct detection of this process is ambiguous. To untangle the initial charge separation mechanism, various mutations have been introduced by multiple research groups to tweak the electronic properties of individual RC cofactors [42,46,47,48,49]. Specifically, mutations targeting the A_0_ cofactor are the focus of current work.

In *Chlamydomonas reinhardtii*, the A_0_ Met → His axial ligand mutants have been studied most extensively. In an ultrafast optical study in the red, long-lived difference spectra observed for the PsaA-M688H and PsaB-M668H mutants were assigned to (A_0A_^−^–A_0A_) and (A_0B_^−^–A_0B_) difference signals, respectively, and it was suggested that forward ET beyond A_0_ was either blocked or slowed in the branch carrying the mutation [50]. Note that PsaA-M688 and PsaB-M668 in *Chlamydomonas reinhardtii* correspond to PsaA-M684 and PsaB-M659 in *Synechocystis* sp. PCC 6803, shown in Figure 1. The amplitudes of the (A_0A_^−^–A_0A_) and (A_0B_^−^–A_0B_) difference spectra were nearly identical in the two mutants, suggesting roughly equal involvement of both branches in ET. A subsequent study of kinetic changes in absorption at 390 nm showed that the formation of phyllosemiquinone (A_1_^−^) in both mutants decreased to about half of that in the wild type (WT), implying that ET was blocked between A_0_ and A_1_ in the affected branch [51]. An EPR study of the PsaA-M688H mutant at 265 K showed the absence of an electron spin polarized (ESP) signal, suggesting that the P700^+^A_1A_^−^ radical pair cannot be formed and that the B-branch transfer, if present, does not produce an ESP signal [52]. In a more recent study in deuterated whole cells of the PsaA-M688H mutant, a spin-polarized spectrum was detected at low temperature (100 K) and assigned to the radical pair P700^+^A_1B_^−^ [53]. The pulsed EPR studies revealed that in the presence of reduced F_X_, the decay of the out-of-phase spin-polarized signal in the WT was biphasic, while it was monophasic in PsaA-M688H or PsaB-M668H, with lifetimes of ~3 μs and ~17 μs, respectively [52,54]. The data were consistent with a blockage of ET from A_0_^−^ to A_1_ in the Met → His mutants. The A_0_ ligand mutants from *Chlamydomonas* were also investigated using femtosecond laser pump-probe studies, where both branch mutants showed an additional bleaching with a maximum at ~681 nm [50]. The time-resolved fluorescence studies with a 3 ps temporal resolution using the PS I samples isolated from the WT and Met → His or Ser mutants supported the model in which P700 is not the primary electron donor with the primary charge separation event occurring between A_−1_ and A_0_ [43].

The A_0_ ligand mutants of Met to Leu, Asn, and His of *Synechocystis* sp. PCC 6803 were generated and characterized [49,55,56,57,58,59,60,61]. It was demonstrated that His could provide a ligand to A_0_ in the Met → His substitutions [60]. The X-band spin-polarized transient EPR spectra of PS I trimers isolated from the A_0_ Met → His mutants at 80 K showed that P700^+^A_1A_^−^ radical pair of PsaA-M684H differed from the WT and PsaB-M659H, suggesting that PsaA-M684H provides an additional hydrogen bond to A_1A_ [49,60]. Interestingly, the X-band spin-polarized transient EPR spectra of the WT and PsaB-M664H were very similar. The room-temperature transient EPR spectra demonstrated that the ET was blocked at A_1A_, and electrons could not be transferred to F_X_ in the A-branch in the PS I complex of PsaA-M684H. This mutant was able to grow photoautotrophically under normal light conditions, although at a slightly slower rate than WT. It is still unclear why PsaB-M659H does not show any obvious difference in EPR spectra and growth rate [49].

In *Chlamydomonas reinhardtii*, two hydrogen bond mutants (Tyr → Phe) to A_0_ have been generated [42,62]. The transient optical absorption spectroscopy and transient electron paramagnetic resonance data showed that the mutations affect the relative amplitudes, but not the lifetimes of the fast and slow phases representing ET from A_1_ to F_X_. Specifically, PsaA-Y696F increased the fraction of the faster component at the expense of the slower component, with the opposite effect observed in PsaB-Y676F. Further study indicated that the rate of primary charge separation was lowered in both mutants, and it was proposed that the primary ET event occurs within an A_−1_/A_0_ pair [42]. However, to date, A_0_ hydrogen bond mutants of *Synechocystis* sp. PCC 6803 comparable to those in *Chlamydomonas reinhardtii* have not been reported.

In this paper, we describe the generation and characterization of three single site-directed variants of PsaA-Y692F, PsaB-Y667A, and PsaB-Y667F and one double site-directed mutation of PsaA-Y692F/PsaB-Y667F in *Synechocystis* sp. PCC 6803, targeting peripheral hydrogen bonds to A_0_ Chl *a* molecules (Figure 2). Note that an analogous double mutant has not been reported in *Chlamydomonas reinhardtii*. The phenotypes of these mutants demonstrate that a nonconservative mutation of an aromatic amino acid to a non-aromatic amino acid affects PS I assembly and/or function. Contrary to expectation, the removal of hydrogen bonds to Chl molecules in Y → F mutants had a minor effect on normal cell growth rate and photosynthetic activity. Similarly, spectroscopic studies showed that these mutations have minimal effect on difference spectra, (P700^+^–P700) and (A_0_^—^A_0_), and have negligible effect on ET kinetics to A_0_ and on overall efficiency of ET in PS I. In contrast, the substitution of tyrosine to alanine, which also interrupts the hydrogen bond to A_0_, was found to be fatal for the organism’s survival, suggesting that this hydrogen bond itself is not critical for the function of PS I, and the primary role of this residue is structural.

## 2. Results

### 2.1. Generation and Characterization of the *Δ*psaAB Recipient Strain

The strategy for generating the recipient strain, Δ*psaAB*, is shown in Figure 3a. The correctness of Δ*psaAB* was confirmed by PCR using the same primer pair from the knockout mutant (Figure 3b). The phenotype of Δ*psaAB* was visually dark blue, which is different from the green color of the WT and RWT strains (Appendix A). The Δ*psaAB* strain could not grow under the normal light intensity (40 μE m^−2^ s^−1^). The fluorescence spectrum of the whole mutated cells at 77 K lacked the characteristic PS I emission band at 730 nm, indicating that PS I expression in Δ*psaAB* was highly suppressed or nonexistent. In contrast, the fluorescence spectrum of the pIBC-transformed RWT strain was indistinguishable from the WT (Appendix A). The phenotype of Δ*psaAB* is similar to that of the recipient strain, Δ*psaB* [63].

### 2.2. Generation of PsaA-Y692F, PsaB-Y667A, PsaB-Y667F, and PsaA-Y672F/PsaB-Y667F

The mutated plasmid was generated based on a PCR method using pIBC as the template for PsaA side mutation and using pBC as the template for PsaB side mutation similar to those described previously [63,64,65]. The plasmid-containing sites were sequenced to ensure that the desired mutations were correct. The double mutation was generated by combining the two single mutations using two restriction enzymes: *Xho* I and *Xba* I and one DNA ligase. To further ensure that the desired PsaA and PsaB mutations were introduced into the Δ*psaAB* and Δ*psaB* recipient strains, respectively, genomic DNAs from a single transformant of a mutant strain were isolated, and the PCR-amplified DNA fragments were directly sequenced. The double mutation was introduced in Δ*psaAB*. The DNA sequencing results confirmed the presence of the designed PsaA-Y692F, PsaB-Y667F, and PsaA-Y672F/PsaB-Y667F mutations.

### 2.3. Physiological Characterizations of PsaB-Y667A: Dramatic Effect

The alanine side chain cannot participate in hydrogen bond interactions, and substituting PsaB-Y667A should disrupt the hydrogen bond between A_0_ and polypeptide. In addition, alanine has a propensity to form alpha helices but can also occur in beta sheets and is generally equivalent to simply truncating a side chain back to the beta carbon. This substitution is unlikely to disrupt the secondary structures of a protein. To examine bacterial growth rate changes caused by the Y to A mutation, PsaB-Y667A was transferred to BG-11 plates without glucose. No visible colonies of PsaB-Y667A were formed on the plate, whereas the RWT cells could form colonies. To further confirm this result, RWT and PsaB-Y667A were cultured under photoautotrophic and photoheterotrophic growth conditions under different light intensities. Under low light intensity (2–3 μmoles m^−2^ s^−1^) and with glucose in the medium, PsaB-Y667A grew slower than the RWT (Appendix A). Under these conditions, heterotrophy was the primary energy acquisition mode, and respiratory electron transport in the variant strain was clearly capable of sustaining heterotrophic growth. PsaB-Y667A could not grow at light intensities that exceeded 40 μmoles m^−2^ s^−1^, even when glucose was present in the medium (Appendix A). Therefore, replacing PsaB-Y667 with a non-aromatic residue resulted in a loss of photoautotrophic growth ability and destructive light sensitivity. No PS I trimers were found in PsaB-Y667A, which prevented further spectroscopic characterizations of the PsaB-Y667A variant PS I.

### 2.4. Physiological Characterizations of PsaA-Y692F, PsaB-Y667F, and PsaA-Y672F/PsaB-Y667F Show Weak Effect of Y → F Mutation

The dramatic effect of the PsaB-Y667A mutation suggests that an aromatic residue may be essential in position 667 of PsaB. Indeed, the substitution of PsaB-Y667 with phenylalanine turned out to have a minor effect on the photosynthetic function of PS I. Both tyrosine and phenylalanine have similar structures. However, the side chain of phenylalanine could not form a hydrogen bond with A_0_ (Figure 2c,d). As stated earlier, two single A_0_ hydrogen bond variants: one on the PsaA side (PsaA-Y692F) and the other on PsaB side (PsaB-Y667F) and a double variant, PsaA-Y692F/PsaB-Y667F, were generated. All these variants were able to grow, showing growth rates similar to the WT in both photoautotrophic and photoheterotrophic conditions, except for the double mutant that showed a slightly slower doubling time under photoautotrophic conditions (Figure 4a). Similarly, the photosynthetic activities of the two single-side mutants reflected in O_2_ production rate were similar to the WT, with the double variant having only slightly lower photosynthetic activity (*t*-test: *p* < 0.05) (Figure 4b). All three variants had Chl and carotenoid contents similar to the WT (Appendix A). Note that while the double variant showed a slightly higher carotenoid level, the difference was within the uncertainty of measurement (*p* = 0.65).

### 2.5. Spectroscopic Characterizations Show Minor Effect of Y → F Mutations

For spectroscopic studies, PS I trimers were isolated and purified (Appendix A). Fluorescence spectra of 77 K indicate a slightly reduced PS I/PS II ratio in the A-branch and double mutants (Appendix A). The electronic absorption spectra of the three variant PS I complexes in the Chl *a* Q_y_ spectral region (600–720 nm) at room temperature is nearly superimposable upon that of the WT PS I complex, with maximum differences being ≤2% (Figure 5). Since optical spectra of P700 and A_0_ in mutants appear to be very similar to the WT (described later), these minor differences in overall absorption spectra of PS I may stem from cumulative changes in numerous antenna Chls affected by long-range structural effects of mutations.

The mutations also have a minor effect on the (P700^+^–P700) spectra (Figure 6). The largest visible difference is observed in the positive band intensity at ~690 nm. This is expected since the major contributor to the (P700^+^–P700) is the special pair P700. The mutations near A_0_ can affect this spectrum in two ways. First, the mutation at this site can affect the electronic properties of a distant Chl either via changes in the electrical field (electrochromic effect) or long-distance structural changes. Such a distant effect has been recently reported for another pigment–protein complex [66], where mutation near BChl *a* in one pocket was shown to affect the electronic properties of BChls in distant pockets. On the other hand, all six Chl *a* molecules in PS I RC are excitonically coupled, the electronic excitation is delocalized over all six pigments, and thus, the (P700^+^–P700) spectrum must naturally reflect the properties of all these pigments as well as excitonic couplings between them.

The transient (A_0_^—^A_0_) absorption difference spectra are shown in Figure 7 (see Appendix A for details). The Gaussian fits to these bands show that the mutational effect on A_0_ absorption is negligible, with a maximum (blue) shift of ~1 nm observed for the A-side mutant. Note that these measurements cannot distinguish between A- and B-branch A_0_ spectra and reflect the superposition of the two branches. Therefore, the 1 nm shift may also be caused by the redistribution of ET between two branches, with A_0_ in each branch having a slightly different absorption spectrum. In either case, the effects of mutations on the electronic Q_y_ transition of A_0_ are negligible.

### 2.6. Energy Transfer and Primary Charge Separation Are Not Affected by Y → F Mutations

The excitation energy transfer was characterized by probing Δ*A* kinetics at 685 nm after exciting into the blue edge of PS I antenna absorption spectrum at 660 nm (Appendix A). The differences in these kinetics are minor; in all cases, the kinetics were well described with two main decay components (28–30 ps and 2.2–2.8 ps) and weaker non-decaying components (in the time window 200 ps) (see Appendix A). The 28–30 ps component is conventionally assigned to overall effective excitation trapping time by the RC, while the 2.2–2.8 ps corresponds to excitation energy equilibration within antenna pigments and is primarily defined by a small number of red-most pigments [5]. It is thus concluded that the Y → F mutation that disrupts the hydrogen bond to A_0_ does not affect RC trap efficiency.

The kinetics of ET from A_0_^−^ to phylloquinone A_1_ was characterized by exciting the PS I complexes at 660 nm and probing absorption changes at 390 nm (Figure 8). It has been shown that the P700^+^ state has a minor contribution at this wavelength, and the positive Δ*A* signal primarily stems from the reduced phylloquinone [34,62,67]. Initially negative Δ*A* signal (photobleaching) stems from the excited states of Chl *a* molecules created by excitation; it decays in ~30 ps, which is consistent with excitation trapping time measured at 685 nm (Appendix A). The non-decaying positive Δ*A* signal in that time window of 600 ps is thus ascribed to the formation of A_1_^−^, indicating that overall ET times are not visibly affected in mutants. The amplitude of this positive component is about the same for WT and single-side mutants, suggesting that the quantum efficiency of light conversion to charge transfer state is also similar, with the double mutant displaying only slightly reduced efficiency, but still within the noise level.

## 3. Discussion

The physical properties of A_0_, such as the absorption spectrum and its very low midpoint potential (−1000–−1100 mV), are primarily shaped by the interactions between the Chl molecule and its protein environment [68]. Any change in these interactions can potentially alter the biophysical properties of A_0_. It has been shown, for example, that the alteration of hydrogen bonding in *Rhodobacter sphaeroides* affects the resonance Raman spectra of the primary electron donor, a dimer of bacteriochlorophyll and bacteriopheophytin, and there is a correlation between multiple hydrogen bonding and midpoint potential of the primary electron donor [69]. Similarly, the midpoint potential and functional properties of the P700 special pair are modulated by the nature of the residues that provide axial ligands to the Mg^2+^ Chl atoms [70].

There are two major interactions between A_0_ and its protein environment [2]. First, the sulfur atoms of methionine residues PsaA-M684 and PsaB-M659 provide ligands to the Mg^2+^ atoms of A_0_ Chl molecules (Figure 1). Second, the keto oxygen of the ring of A_0_ Chl molecule is hydrogen-bonded by the hydroxyl groups of the tyrosine residues PsaA-Y692 and PsaB-667 in branch A and B, respectively (Figure 2a,b). The Y → F replacement of the latter residues removes these hydrogen bonds (Figure 2c,d). The associated changes in a midpoint potential of A_0_ are expected to cause a change in the primary ET rates, as well as in the transfer rate from A_0_ to A_1_. Significant changes in the primary charge separation rate would affect excitation trapping efficiency and reflect in the overall lifetime of electronic excitation in the antenna. Changes in ET rate from A_0_ to A_1_ would lead to changes in the fraction of accumulated P700^+^A_0_^−^ state, and, if sufficiently slow (longer than excitation trapping time), an additional slower decay component could be detected in ultrafast pump-probe signals that would correspond to the decay of A_0_^−^ state. None of this was observed in Y → F mutants of A_0_ pockets. The Moser–Dutton model [71,72] predicts that an A_0_ redox potential change of ~0.05 eV or larger would be resolved in the kinetic measurements, provided that the donor–acceptor distance and reorganization energy are not altered. Curiously, the present results imply that the hydrogen bond to the A_0_ in both branches has a minor, if any, effect on its midpoint potential. This is consistent with the findings in [42,62], where the analogous substitution was studied separately in each of the RC branches of *Chlamydomonas reinhardtii*. On the other hand, the substitution of Tyr with a smaller non-aromatic Ala residue (PsaB-Y667A) is fatal for the organism. The structural effect of this mutation could be caused by the proximity of this residue to PsaB-W668, which has a π-π interaction with phylloquinone. It strongly suggests that the primary role of these tyrosine residues is structural, and their ability to provide hydrogen bond ligands to Chls is not critical for its function. The aromatic residues, such as Tyr and Phe, are, however, required to maintain an optimal local structure for A_0_ binding.

It is also found that Y → F substitution has an insignificant effect on the electronic absorption of A_0_ pigment (<1 nm). Substitutions near the Chl *a* keto group have been explored in detail in the mutagenesis study of the LvWSCP protein [73]. Measurements of the absorption spectra supported by classical modeling demonstrated that Chl *a* site energies are red-shifted if a polar H-bonded residue at the keto group is mutated to a positively charged one. However, polar-to-hydrophobic mutation (Glu to Ala) in [73] resulted in only 0.1 nm blue shift. Our electronic structure calculations of Chl *a* with Tyr and Phe residues in the keto position suggest that neither Tyr nor Phe produces a sizeable effect on the Chl *a* Q_y_ transition, with a predicted shift due to Y → F mutation of 15 cm^−1^ or ~0.7 nm (see Appendix A). Electronic detachment and attachment densities of the Chl *a* Q_y_ transition, computed at the TDDFT wB97x-d/6-31+G* level of theory, are shown in Figure 9. The detachment density corresponds to the electron density vacated in the electronic transition, while the attachment density is the density becoming occupied in the transition. In other words, differences between detachment and attachment densities visualize the redistribution of the electronic density (or electron charge) upon excitation. As Figure 9 reveals, the attachment (blue) density is slightly bigger than the detachment (red) density at the keto oxygen, suggesting that a small amount of electronic charge (~−0.002 e) is accumulated there in the excited state. Generally, the accumulation of the electronic charge in the excited state on the oxygen that donates its density to the hydrogen bond should stabilize (red-shift) the excited state, such that breaking this hydrogen bond should produce the opposite effect of destabilizing the excited state. However, as both experiments and calculations suggest, the effect is minor and can easily be counteracted by other effects, such as interactions with π cloud of neighboring residues, steric strain, and structural changes due to mutation. It is noteworthy, though, that a similar disruption of the peripheral hydrogen bond to the BChl *a* keto group in the Y345F mutant of the Fenna Mathews Olson antenna complex led to a blue-shift of the BChl Q_y_ absorption band up to 4 to 8 nm [66,74]. However, our electronic structure calculations suggest that the Q_y_ charge accumulation at BChl *a* keto group is larger (−0.007 electron) than at the corresponding group in Chl *a* (−0.002 electron), in accordance with the more substantial effect observed in BChl *a*.

While here we did not have capabilities to perform nanosecond kinetic measurements to quantify the effect of mutations on the ratio of ET along A- and B-branches, it is expected that it would be similar to the results obtained for analogous single-branch Tyr → Phe substitutions in *Chlamydomonas reinhardtii* [42,62]. Based on the effect of mutations on branching ratio, it was concluded that the primary electron donor is A_−1_ and not P700, and, thus, the initial charge separation state is A_−1_^+^A_0_^−^, and the P700^+^A_0_^−^ is formed in a second ET step from P700 to A_−1_^−^ [62] This could be, however, an oversimplified picture, as RC Chls are strongly coupled [75] and may act as one supermolecule. For example, P700 and the two A_−1_ pigments could form a symmetric exciplex in which the excited state, excitonically delocalized over these four pigments, is mixed with two charge transfer states, P700^+^ A_−1A_^−^ and P700^+^ A_−1B_, as proposed in [46,47]. The first ET then occurs directly from this exciplex to A_0_. Any asymmetric changes in the electronic properties of A_0_ Chls can then tilt ET to prefer one branch or the other. It cannot be also excluded that the charge transfer character of excitonic excitation delocalized over all six RC Chls promotes the formation of P700^+^A_0_^−^ in a single step. The assignment of A_−1_ as a primary electron donor was also inferred in [42]. However, this study used a compartmental energy transfer scheme to model the results of the optical pump-probe experiment. In this approach, most of the antenna Chls were modeled as a single entity with separate compartments for P700 and for each of the remaining RC Chls. Such a model ignores rich energy transfer processes within the antenna and unavoidably assigns respective kinetic signals to ET events, making results ambiguous.

## 4. Materials and Methods

### 4.1. Generation of a New *Δ*psaAB Recipient Strain of Synechocystis *sp.* PCC 6803

A new recipient strain was generated by the deletion of a 3′-end portion of the *psaA* gene and the whole *psaB* gene as described before [63,64] with minor changes. Generation of the new *psaAB* knockout (Δ*psaAB*) was performed by cloning a portion of the *psaA* ORF between 764 bp and 1320 bp (the whole gene length is 2256 bp) with a *Bam*H I restriction site introduced at its 3′ terminus via polymerase chain reaction (PCR). A portion of DNA downstream of the *psaB* ORF was also cloned with the introduction of a *Bam*H I restriction site at its 5′ terminus via PCR. These two amplified DNA fragments were purified from agarose gel prior to being inserted into pGEM-T vector (Promega, Madison, WI, USA) by ligation. A spectinomycin resistance cassette (Sp^R^) was inserted between the two flanking fragments using the unique *Bam*H I site. The resulting plasmid was called pGEM-T-Δ*psaAB*. pGEM-T-Δ*psaAB* was transformed into the *Synechocystis* sp. PCC 6803. A stable Δ*psaAB* strain was obtained via homologous recombination after 9 consecutive passages on the culture medium containing 20 µg/mL spectinomycin. A Δ*psaB* recipient strain was generated by the deletion of a 3′-end portion of the *psaB* gene as described before [64] with minor changes [63]. The *psaB* knockout strain (Δ*psaB*) was obtained by cloning a 3′-end portion of the *psaB* ORF (from 608 bp to 1014 bp) and a fragment 9 to 751 bp downstream of *psaB*. An *Eco*R I restriction site was created at the 3′-end of the first fragment and 5′-end of the second fragment. A kanamycin resistance cassette (Km^R^) was inserted into the *Eco*R I restriction site. The resulting plasmid was named pGEM-T-Δ*psaB*. This plasmid was transformed into the *Synechocystis* sp. PCC 6803. A stable Δ*psaB* strain was obtained after 9 consecutive passages on the culture medium containing 20 μg/mL kanamycin. Both stable Δ*psaAB* and Δ*psaB* strains were verified by PCR.

### 4.2. Generation of the Site-Directed Variants and Transformation of the Designed Mutations into the *Δ*psaAB and *Δ*psaB Recipient Strains

The pIBC and pBC plasmids [64] served as the templates for generating PsaA and PsaB mutants, respectively, using Q5 Site-Directed Mutagenesis Kit (New England BioLabs Inc., Ipswich, MA, USA). The site-directed mutated DNAs of *psaA* were used to transform the recipient strain Δ*psaAB*, and the site-directed mutant DNAs of *psaB* were used to transform the recipient strain Δ*psaB*. The transformation and transformant selection were performed under low light (2–3 μmoles m^−2^ s^−1^) and heterotrophic growth using 10 µg/mL chloramphenicol as described in reference [64]. The chloramphenicol resistant colonies were segregated for at least four generations through single colony selection each generation. After complete segregation, the genomic DNAs were isolated from the mutants. The DNA fragments harboring the designed mutations were amplified by PCR, and the PCR products were sequenced to verify the correctness of the mutants. The pIBC and pBC plasmids were introduced back into the Δ*psaAB* and Δ*psaB* recipient strains to generate the recovered wild types (RWTs), respectively, which served as positive controls.

### 4.3. Genomic DNAs Preparation of Synechocystis *sp.* PCC 6803

*Synechocystis* sp. PCC 6803 cells were grown to around 2.0 OD_730_ in the BG-11 medium and harvested by centrifugation at 4500 rpm for 5 min. (Eppendorf 5415C, Hamburg, Germany). The cells were resuspended in 400 µL of the TE buffer (10 mM TrisHCl, 1 mM EDTA, pH 7.0), and 8 µL of 10% SDS, 16 µL 5% lauryl sarcosine, 200 µL of autoclaved glass beads, and 400 µL of TE saturated phenol were added. After vertexing 3 times for 30 s and centrifugation at 13,000 rpm for 10 min (Eppendorf 5415C, Hamburg, Germany), the top layer was collected and extracted consecutively with 400 µL phenol, 400 µL phenol/chloroform (1:1), and then, 400 µL chloroform. After addition of 1/10 volume of 3.0 M sodium acetate and 2 × volumes of ice-cold ethanol (100%), DNA was precipitated at −20 °C for 20 min. DNA was pelleted by centrifugation at 13,000 rpm for 10 min. The pellet was dissolved in 30 µL of the TE buffer. Rnase was added into the DNA solution, and DNA solution was incubated at 37 °C for 30 min. DNA was precipitated again by addition of 2 × volumes of ice-cold 100% ethanol and incubation at −20 °C for 20 min. RNA-free DNA was collected by centrifugation at 13,000 rpm for 10 min, vacuum dried, and dissolved in 30 µL of dH_2_O.

### 4.4. Growth of the Synechocystis *sp.* PCC 6803 Cells

The WT and the variants of *Synechocystis* sp. PCC 6803 were grown on BG-11 medium agar plates at 30 °C supplemented with 5 mM glucose, 10 mM TES-NaOH (pH 8.0), and 0.3% sodium thiosulfate. The appropriate antibiotics (10 µg/mL chloramphenicol or 20 µg/mL kanamycin or 20 µg/mL spectinomycin) depending on the recipient strains and the site-directed variants were added in the agar plates. The variants were maintained in the illuminated incubators (MODEL 818, ThermoFisher Scientific Inc., Waltham, MA, USA) under a 8 h-light/16 h-dark photoperiod cycle. Different light intensities were chosen for growing each strain: The Δ*psaAB* and Δ*psaB* recipient strains were grown exclusively in the dark with only 15 min of illumination with 40 µE m^−2^ s^−1^ light intensity per day. Liquid cultures were grown photoautotrophically or photomixotrophically (with 5 mM glucose) in BG-11 liquid medium with appropriate antibiotics by shaking at 130 rpm or by bubbling with filtered air as previously described [63].

### 4.5. Isolation of Thylakoid Membranes and Purification of PS I Complexes

The *Synechocystis* sp. PCC6803 cells were harvested during the late exponential growth by centrifuging at 4000× *g* for 6 min and washed once using 10 mM Mops-NaOH, pH 7.0 buffer. The cells were resuspended in SMN buffer (0.4 M sucrose, 10 mM NaCl, and 10 mM Mops-NaOH, pH 7.0). The protease inhibitor, phenylmethanesulfonyl fluoride, was added to the cultures to a final concentration of 0.2 mM. The harvested cells were broken at 4 °C by a bead-beater (Cat. No. 909, BioSpec Products Inc., Bartlesville, OK, USA) using a 50 mL chamber and 0.1 mm glass beads. The procedure included twenty consecutive cycles with 1 min breaking and 1 min resting. Debris were discarded via centrifugation at 4000× *g* for 30 min. Remaining supernatants were then pelleted by Sorvall LYNX4000 centrifugation at 13,500 rpm (ThermoFisher Scientific Inc., Waltham, MA, USA) for 1 h. The thylakoid membranes were resuspended in 10 mM Mops-NaOH, pH 7.0, and homogenized to ensure a uniform and even mixture of the membranes.

To purify the PS I complexes, thylakoid membranes were diluted to a concentration of 0.5 mg/mL Chl in 10 mM Mops-NaOH, pH 7.0 buffer and solubilized for 30 min at 4 °C by adding n-dodecyl-D-maltopyranoside (DM) to a concentration of 1.5% (*w*/*v*). The solution was centrifuged for 20 min at 13,500 rpm to remove insoluble debris, and the supernatant was loaded onto a 10–30% (*w*/*v*) linear sucrose gradient prepared in 10 mM Mops-NaOH pH 7.0 buffer containing 0.05% (*w*/*v*) DM. The gradients were centrifuged for 18 h at 29,600 rpm at 4 °C (Beckman Coulter Optima XE Ultra Centrifuge, SW32 Ti rotor, Brea, CA, USA). The lower green band containing PS I trimers was collected. The extracted PS I trimers were then concentrated by using Pierce concentrators using 30,000 Dalton molecular weight cutoff membranes (ThermoFisher Scientific Inc., Waltham, MA, USA) and either used immediately or stored at −80 °C for later studies.

### 4.6. Measurement of Cell Growth Rate and Quantitation of Chlorophyll and Carotenoid Content

The cells at a late exponential growth phase were harvested and centrifuged at 4400× *g*, and the pellet was suspended in BG-11 medium without glucose. This procedure was repeated three times to remove glucose from the previous cell culture. The cell cultures were then resuspended in BG-11 medium with or without glucose for photomixotrophic or photoautotrophic growth, respectively, under 40 µE m^−2^ s^−1^ light intensity, 8 h-light/16 h-dark photoperiod cycle at 30 °C, while shaken constantly at 130 rpm. The cell density was monitored by the absorbance at 730 nm (A730) every day for 7–10 days. At this wavelength, pigments involved in photosynthesis have negligible absorbance, and the signal is dominated by light scattering by the cells and, thus, is proportional to the cell concentration. To quantify pigment contents, the number of cells in the extract was estimated using A730 absorption, and the contents of Chl and carotenoid molecules extracted from the cells using 100% methanol were determined spectrophotometrically [76].

### 4.7. Oxygen Evolution Measurements

Oxygen evolution measurements of the WT and the variant cells were performed by using a Chlorolab-2 oxygen electrode under 900 µE m^−2^ s^−1^ actinic light intensity derived from LH36/2R light source (Hansatech, Norfolk, UK). The temperature of the measuring chamber was maintained at 25 °C, and the cell concentrations were adjusted to absorbance A730 = 2 per mL (2 OD_730_/_mL_) in 25 mM HEPES-NaOH, pH 7.0 buffer. CO_2_ was provided to the cells through a supplement of 10 mM NaHCO_3_.

### 4.8. UV Absorption Spectra of PS I Complexes

The room-temperature absorption spectra were measured using Agilent 8453 UV-Vis spectrophotometer. The purified PS I samples with 10 μg of Chl were diluted with 10 mM MOPS-NaOH buffer, pH 7.0, to a final volume of 1.0 mL.

### 4.9. 77 K Fluorescence Spectroscopy of Whole Cells

The fluorescence emission spectra of whole cells were measured using the protocol described in [28] with minor changes. Cells with 2.0 OD at 730 nm were harvested during exponential growth phase and resuspended in 25 mM HEPES-NaOH, pH 7.0 buffer. Glycerol was added to a final concentration 70% (*v*/*v*). The final volume of the samples for 77 K fluorescence spectroscopy was 100 µL. Samples were incubated in the dark on ice for at least 5 min, frozen to 77 K, and fluorescence spectra were measured by a Cary Eclipse fluorescence spectrophotometer using excitation wavelength 440 nm with both excitation and emission bandwidths set to 10 nm.

### 4.10. (P700^+^–P700) Difference Spectra

The method for measuring the (P700^+^–P700) difference spectra has been described in detail earlier [63,77]. Upon excitation, a charge-separated state P700^+^[F_A_/F_B_]^−^, is formed within an ms, followed by back-recombination and formation of a neutral P700 (open RC) within ~50–100 ms. However, a small fraction of electrons on terminal acceptors F_A_/F_B_ are scavenged by oxygen, resulting in a long-lived P700^+^ state [78,79] (closed RC) that can persist for hours. External electron donors can shorten this lifetime by orders of magnitude down to seconds. Thus, prolonged light flashes can reversibly oxidize nearly all P700 in a PS I sample by exciting PS I over and over again, “closing” the RCs. In this work, (P700^+^–P700) spectra were measured at room temperature in samples of PS I trimers containing 100 mM sodium L-ascorbate (electron donor), housed in 10 mm pathlength optical cells (OD ~1.0 at 680 nm). A Varian Cary 300 Bio UV-Vis spectrophotometer was used to measure absorption profiles for samples with open and closed RCs. After a ~1 min period of darkness, several times longer than needed for sodium ascorbate to reduce all P700^+^ to a neutral P700 state, an absorption was recorded at a given probe wavelength (P700 spectrum). A 20 mW blue LED then illuminated the sample for 4 s, closing all RCs by producing P700^+^. The absorbance at the same wavelength was then measured again 2 s after the LED was switched off and averaged over the next 4 s. The procedure was repeated for all wavelengths resulting in the P700^+^–P700 difference spectrum. P700^+^ reduction dynamics were also recorded for each sample, and the corresponding kinetic rates were used to calculate the absorption change at t = 0, i.e., immediately after the LED was switched off, so that P700^+^–P700 spectral intensities measured for different samples could be directly compared.

### 4.11. Ultrafast Time-Resolved Optical Spectroscopy

The optical pump-probe spectrometer is described in detail elsewhere [77]. Briefly, femtosecond pump and probe pulses were delivered from a home-built laser system comprising the optical parametric amplifier (OPA) pumped by a regeneratively amplified self-mode-locked Ti:sapphire laser system operating at a 1 kHz repetition rate. The OPA output was tuned to 660 nm and used as a source of the pump pulses. Transient absorption changes Δ*A* in samples were probed in the 665–720 nm range using broadband white light continuum pulses generated by passing an amplified Ti:sapphire laser output through a 1 mm thick sapphire plate. For experiments probing Δ*A* near 390 nm, the sapphire plate was replaced with a frequency-doubling crystal to generate a second harmonic of the 780 nm fundamental beam. Pump and probe pulse durations were ~200 fs. Pump and probe pulse polarizations were set at a magic angle (54.7°) to each other to avoid depolarization effects due to energy transfer. An Oriel MS257 imaging monochromator operating at ≤3 nm bandwidth dispersed the probe and reference light onto two Hamamatsu S3071 Si pin photodiodes; the signals were collected and processed by an automated computer system.

The pump pulse intensity was set to be sufficiently low to avoid double-excitation of a single PS I complex, which could lead to shorter apparent excitation decays due to excitation annihilation effects [5]. All experiments were carried out at room temperature. PS I concentration in the samples was tuned so that optical density at 660 nm was 0.23 ± 0.03 in a 1 mm pathlength optical cell. To avoid exciting the same spot in the sample by several consequent pump pulses, the cell housing the samples was kept in continuous motion by utilizing a home-built 2-axis motorized translation stage, providing a constant linear translation speed of 16 mm/s.

### 4.12. Measuring (A_0_^—^A_0_) Spectra

Due to the A_0_^−^ lifetime being shorter than the energy transfer time from the antenna to RC, this state never accumulates to a 100% population. The (A_0_^—^A_0_) signature is therefore extracted using the approach described in [80,81] using ultrafast absorption difference measurements. The PS I complexes are excited at 660 nm, which pumps the blue edge of the core antenna spectrum, and the absorption changes Δ*A* are probed in 660 to 720 nm interval at fixed time delays. To isolate absorption changes associated with ET within RC, transient spectra are measured for both closed and open reaction centers. Since antenna lifetime has been shown to be independent of the state of RC and there is no ET in complexes with closed RC, the subtraction of Δ*A* profile measured for closed centers from that measured for open centers eliminates the contribution from energy transfer processes within antenna in the signal, and the resulting ΔΔ*A* signal reflects absorption changes due to ET. These spectra are measured at fixed times after excitation—8 ps and 200 ps. At 200 ps, the energy transfer and primary charge separation are complete with 100% of complexes in P700^+^A_1_^−^ state, while at 8 ps, ET state is a mix of P700^+^A_0_^−^ and P700^+^A_1_^−^ states. Since A_1_ is a phylloquinone with no spectral features at around 700 nm, the difference in spectral shapes at 8 ps and 200 ps, i.e., ΔΔΔ*A* spectrum, extracts the (A_0_^—^A_0_) spectrum.

### 4.13. Electronic Structure Calculations

The initial structure of the Chl-Tyr complex has been obtained from the PDB ID: 5OY0 (Psa a-Y692), truncated at the C3 carbon of a phytol tail, protonated, and optimized with harmonic constraints [82] at the density functional theory (DFT) wB97x-d/6-31G* level of theory [83]. Geometries of the Chl-Phe and Chl *a* complexes have been obtained from the Chl-Tyr geometry without further reoptimization. Excited state calculations of these complexes have been performed at the TDDFT wB97x-d/6-31+G* level of theory in the gas phase. All electronic structure calculations have been performed in the Q-Chem 6.0 electronic structure package [84]. Electron attachment and detachment densities have been plotted using the VMD 1.9.4 visualization software [85].

### 4.14. Statistical Analyses

*t*-test was used to identify statistical differences between the different feature engineering methods’ similarity values. Conventionally, a threshold of *p* < 0.05 was used to determine significance; the differences with *p* > 0.05 were considered insignificant.

## 5. Conclusions

The functional properties of Chl *a* molecule in A_0_ pockets of PS I RC have been targeted by Tyr → Phe and Tyr → Ala substitutions at PsaA-Tyr692 and PsaB-Tyr667 positions. Both substitutions eliminate the peripheral hydrogen bond of the native Tyr to the keto group of Chl *a.* The Tyr → Phe mutations have minimal effects on the electronic properties of A_0_ Chls, as indicated by the negligible effects of respective mutations on absorption spectra of A_0_ and energy/electron transfer kinetics in PS I. These results are supported by DFT modeling of the effect of this hydrogen bond on the electronic structure of Chl *a*. This is surprising, since analogous mutations in pockets of BChl *a* molecules in the Fenna Matthews Olson photosynthetic complex caused significant spectral shifts [66,74]. On the other hand, it is found that the substitution of tyrosine with a much smaller non-aromatic alanine (PsaB-Y667A) is fatal for the photosynthetic function of PS I. It is thus inferred that the hydrogen bonds provided by PsaA-Tyr692 and PsaB-Tyr667 are not critical and that the role of that residue is primarily structural.

Despite considerable experimental data on the kinetics of charge separation in PS I RC mutants targeting properties of electron transfer cofactors, the conclusions on the mechanism of initial charge separation in RC remain ambiguous. One of the promising routes to solve this long-standing question is predictive QM/MM modeling from the first principles, similar to the approach used recently by two co-authors of the current paper to solve the electronic structure of the Fenna Matthews Olson complex and its mutants [66,86]. The results presented here, along with other experimental studies of PS I RC mutants, will serve as a critical gauge for developing and testing QM/MM modeling protocol that will lead to unambiguously revealing the elusive details of charge separation in RC.

## Figures and Tables

**Figure 1 ijms-25-04815-f001:**
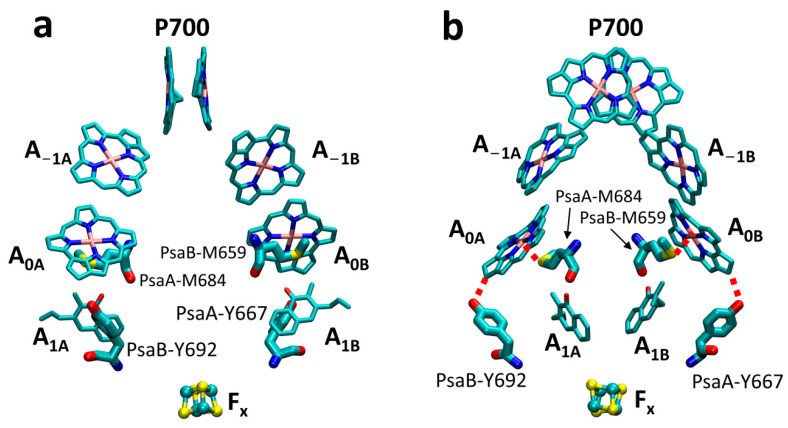
PS I reaction center consists of two pseudo-symmetrical electron transfer branches, A and B, comprising six Chl *a* pigments (pairs P700, A_−1_ and A_0_) and two phylloquinones (A_1_); the two branches converge on iron–sulfur complex F_x_. Also shown are residues PsaA-Y692 and PsaB-Y667 that provide peripheral hydrogen bonds to A_0A_ and A_0B_ cofactors, respectively, and axial ligands PsaA-M684 and PsaB-M659. Panels (**a**,**b**) show two views at different angles to better visualize the structure and location of hydrogen bonds, as shown by red dashed lines in panel (**b**). The structure is taken from PDB ID: 5OY0.

**Figure 2 ijms-25-04815-f002:**
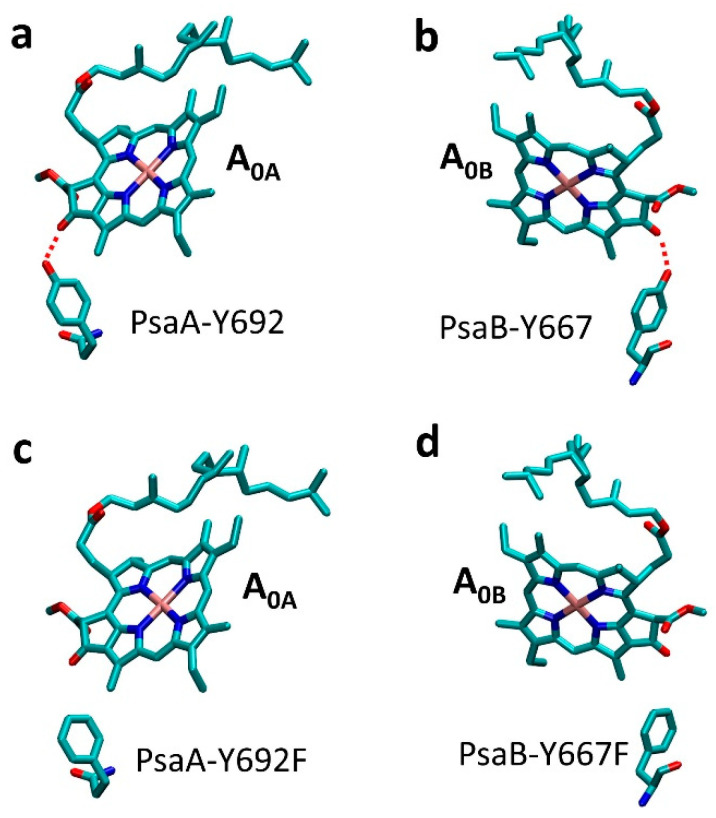
(**a,b**) Local structures of A_0A_ and A_0B_ Chls in WT PS I RC. Red dashed lines show hydrogen bonds to tyrosines. (**c**,**d**) Residues PsaA-Y692 and PsaB-667 were mutated to phenylalanine, disrupting hydrogen bonding to these cofactors.

**Figure 3 ijms-25-04815-f003:**
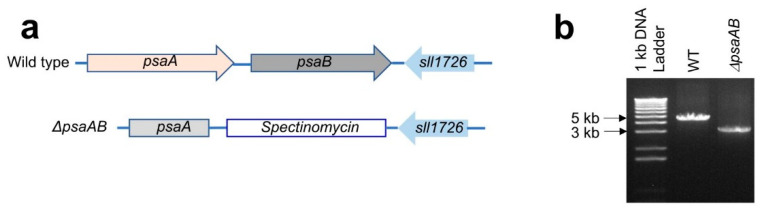
(**a**) The scheme for generating the Δ*psaAB* recipient strain of *Synechocystis* sp. PCC 6803; (**b**) The PCR analysis shows the PCR products from the WT, recipient, and recovered WT stains by using the same relatively primers to generate the knockout strain.

**Figure 4 ijms-25-04815-f004:**
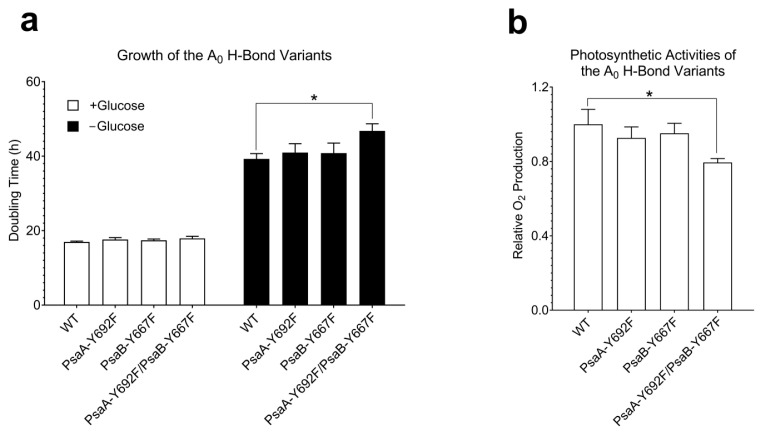
(**a**) Growth rates (cell doubling times) of WT and mutants in the presence of glucose under low light intensity (2–3 μmoles m^−2^ s^−1^, white bars) and in the absence of glucose under normal light intensity (40 μmoles m^−2^ s^−1^). (**b**) The photosynthetic activities of the two single-side mutants reflected in O_2_ production rate are similar to the WT, with the double variant having only slightly lower photosynthetic activity. * means *p* < 0.05 using *t*-test.

**Figure 5 ijms-25-04815-f005:**
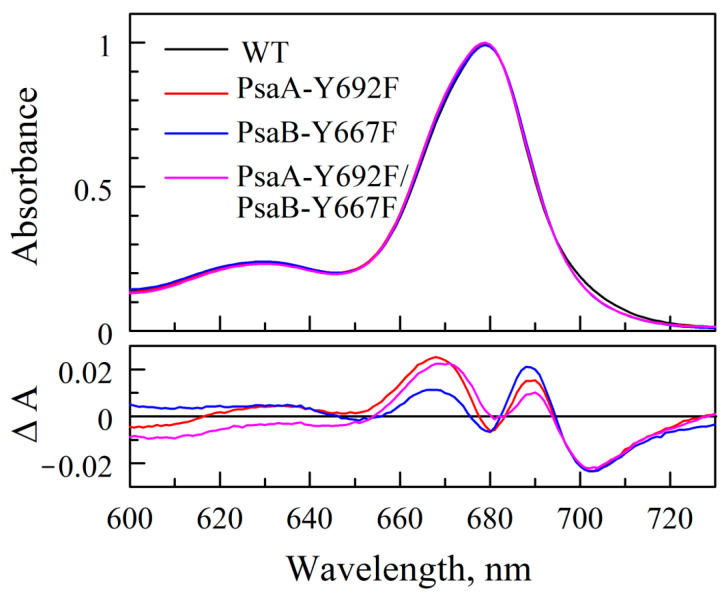
Steady-state absorption spectra of purified PS I complexes from mutants are nearly identical to that of WT (**upper panel**), with differences ≤ 2% (**lower panel**).

**Figure 6 ijms-25-04815-f006:**
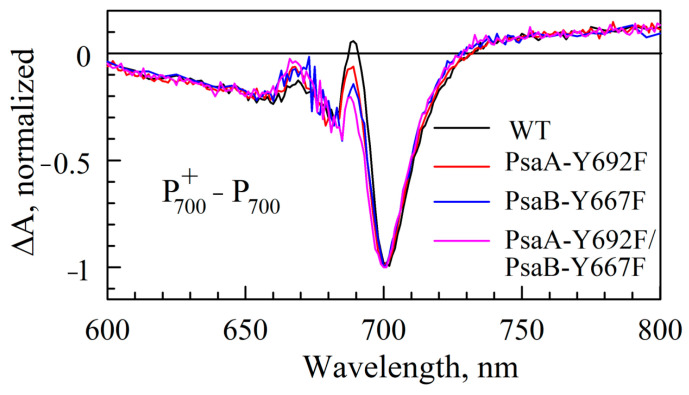
(P700^+^–P700) absorption difference spectra of mutants differ only marginally from that of WT, showing that mutations have minimal effect on the electronic structure of P700.

**Figure 7 ijms-25-04815-f007:**
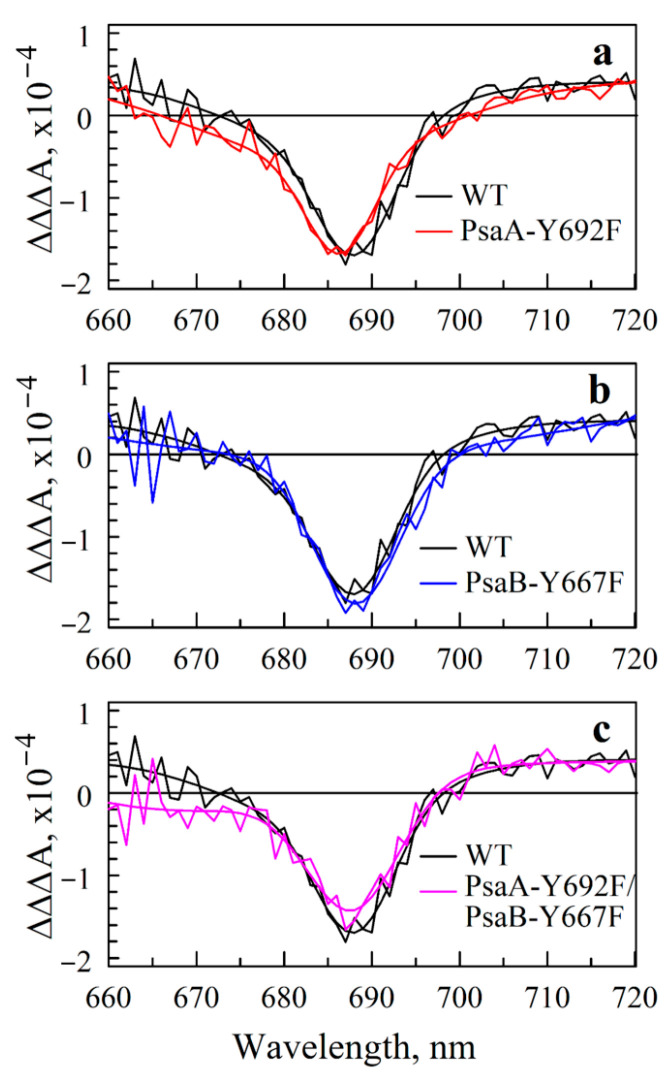
(A_0_^−^–A_0_) spectra of mutants are superimposed on respective WT spectrum, showing that effect of mutations on the absorptive properties of A_0_ is insignificant. The black line is the (A_0_^−^–A_0_) spectrum of WT in all panels; colored lines show spectra measured for PsaA-Y692F (**a**), PsaB-Y667F (**b**), and double mutant PsaA-Y692F/PsaB-Y667F (**c**). Smooth lines are Gaussian fits to respective profiles plotted to ease visual comparison.

**Figure 8 ijms-25-04815-f008:**
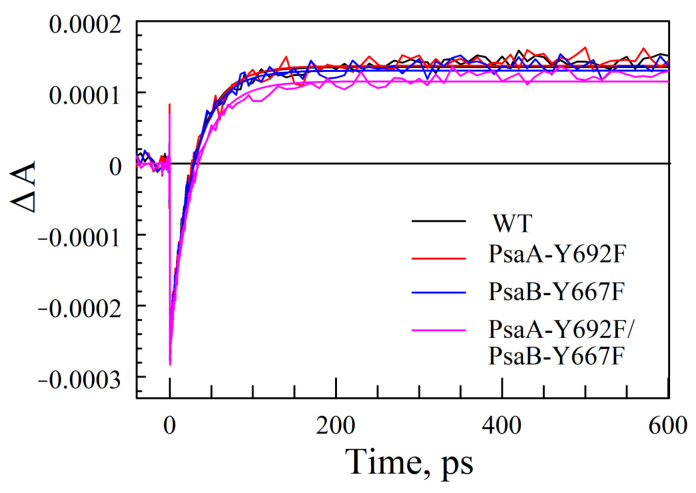
Absorption difference kinetics measured at 390 nm for WT and mutants after exciting complexes at 660 nm. The fast (~30 ps) initial bleaching decay component stems from antenna excitation decay, and its amplitude reflects the number of excitations created by the excitation pulse; the latter is normalized for all curves to WT signal. The long-living positive component stems from the formation of A_1_^−^ showing that the electron transfer efficiency is almost identical in mutants and WT complexes.

**Figure 9 ijms-25-04815-f009:**
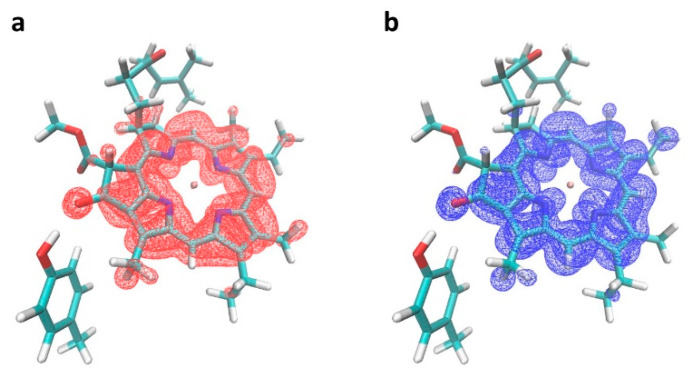
Computed electronic detachment (**a**) and attachment (**b**) densities of the Chl *a* Q_y_ transition, computed at the TDDFT wB97x-d/6-31+G* level of theory, reveal minimal accumulation of electronic charge in the excited state at the keto oxygen, i.e., the attachment (blue) density is slightly bigger than the detachment (red) density at the keto group.

## Data Availability

Data is contained within the article and Appendix A.

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
