# Peer review of "Impact of Peripheral Hydrogen Bond on Electronic Properties of the Primary Acceptor Chlorophyll in the Reaction Center of Photosystem I"

_ijms, 2024, doi:10.3390/ijms25094815_

Round 1
Reviewer 1 Report
Comments and Suggestions for Authors
The manuscript titled “Impact of peripheral hydrogen bond on electronic properties of the primary acceptor chlorophyll in the reaction center of photosystem I” by Luo, L.; et al. is a scientific work where the authors studied the specific contribution of certain residues on the electron transfer mediated by the hydrogen bonding. For it, the performance of four enzyme mutants and two-double knocked-out specimens were tested to visualize the differences compared to the wild-type plastocyanin-ferredoxin oxidoreductase enzyme. The most reventant finding obtained in this research could be interesting for a specialized target audience. Furthermore, the manuscript is generally well-written.
However, it exists some points that need to be addressed (please, see them below detailed point-by-point). The most relevant outcomes remarked by the authors can contribute in the growth of many fields by the better understanding of the underlying mechanisms involved in the electron transfer that takes place in the photosystem I. For this reason, I will recommend the present scientific manuscript for further publication in the International Journal of Molecular Sciences once all the below described suggestions will be properly fixed.
Here, there exists some points that must be covered in order to improve the scientific quality of the manuscript paper:
1) ABSTRACT. “Contrary to expectations (…) DFT modeling (…) Chls” (lines 36-38). Please, the authors should define the full-terms of DFT (density functional theory) and Chls (Chlorophyll). Then, the abbreviations should be placed between brackets. This comment should be taken into account for the rest for the main manuscript body text.
2) KEYWORDS. The authors should consider to add the term “chlorophyll” in the keyword list.
3) INTRODUCTION. This section clearly depicts the state-of-the-art related to the examined field. “Many of these (…) photosystem I (PSI), the light-driven ferredoxin-plastocyanin oxidoreductase of cyanobacteria (…) reaction center (RC) that iniatiates electron transfer processes (…) excitation energy to RC” (lines 49-55). Even if I agree with this statement furnished by the authors, it should not be neglected other biology systems as the ferredoxin-NADP+ reductase (FNR) which has been shown to be directly linked to the electron transfer taken placed in the PSI [1] mediated by iron-sulfur ferredoxin protein in normal conditions or flavodoxin when iron is scarced in the medium [2].
[1] Marco, P.; Elman, T.; Yacoby, I. Binding of ferredoxin NADP+ oxidoreductase (FNR) to plant photosystem I. Biochim. Biophys. Acta Bioenerg. 2019, 1860, 689-698. https://doi.org/10.1016/j.bbabio.2019.07.007.
[2] Marcuello, C.; de Miguel, R.; Martínez-Júlvez, M.; Gómez-Moreno, C.; Lostao, A. Mechanostability of the Single-Electron-Transfer Complexes of Anabaena Ferredoxin-NADP(+) Reductase. Chemphyschem. 2015, 16, 3161-3169. https://doi.org/10.1002/cphc.201500534.
4) “An increasing number of PSI structures of (…) showing a high resemblance between structures” (lines 61-63). Could the authors quantify the degree of resemblance between the aforemention PSI structures among the existing biology sources.
5) METHODS. “To purify the PSI complexes (…) to a concentration of 1.5% (w/v)” (lines 250-252). Did the authors observe any aggregation effects during the purification process? In case affirmative, what strategies were followed in order to minimize this detrimental effect? Some insights should be provided in this regard.
6) “The pump pulse (…) annihilation effects [37][REF]” (lines 333-335). Please, it lacks a reference citation at the end of this statement. The authors should fix this point.
7) Finally, a subsection related to the statistical analysis carried out by the authors should be added in Materials & Methods.
8) RESULTS. This section unequivocally shows the data gathered by the authors in the different techniques used in this research. Did the authors carry out stopped flow measurements (or other analogous experiment) to discern the electron transfer reaction kinetics among the wild-type and the mutants and how the specific changes on the aminoacid residues could affect to the electron catalytic processes? Some information should be provided in this regard.
9) DISCUSSION. This section clearly states the most relevant outcomes found in this research. The authors should consider to add some potential future action lines to pursue this research.
Comments on the Quality of English LanguageThe manuscript is generally well-written albeit it may be desirable if the authors could recheck it in order to polish final details susceptible to be improved.
Author Response
Reviewer #1
The authors appreciate that reviewer #1 took the time to review the manuscript. The comments are insightful and helpful, and addressing the concerns will help improve the manuscript's quality.
Here, there exists some points that must be covered in order to improve the scientific quality of the manuscript paper:
1) ABSTRACT. “Contrary to expectations (…) DFT modeling (…) Chls” (lines 36-38). Please, the authors should define the full-terms of DFT (density functional theory) and Chls (Chlorophyll). Then, the abbreviations should be placed between brackets. This comment should be taken into account for the rest for the main manuscript body text.
We used the full term for DFT and Chls in the abstract. We also carefully checked the usage of full names and abbreviations throughout all other sections of the manuscript.
2) KEYWORDS. The authors should consider to add the term “chlorophyll” in the keyword list.
We have added “chlorophyll” to the keyword list.
3) INTRODUCTION. This section clearly depicts the state-of-the-art related to the examined field. “Many of these (…) photosystem I (PSI), the light-driven ferredoxin-plastocyanin oxidoreductase of cyanobacteria (…) reaction center (RC) that iniatiates electron transfer processes (…) excitation energy to RC” (lines 49-55). Even if I agree with this statement furnished by the authors, it should not be neglected other biology systems as the ferredoxin-NADP+ reductase (FNR) which has been shown to be directly linked to the electron transfer taken placed in the PSI [1] mediated by iron-sulfur ferredoxin protein in normal conditions or flavodoxin when iron is scarced in the medium [2].
[1] Marco, P.; Elman, T.; Yacoby, I. Binding of ferredoxin NADP+ oxidoreductase (FNR) to plant photosystem I. Biochim. Biophys. Acta Bioenerg. 2019, 1860, 689-698. https://doi.org/10.1016/j.bbabio.2019.07.007.
[2] Marcuello, C.; de Miguel, R.; Martínez-Júlvez, M.; Gómez-Moreno, C.; Lostao, A. Mechanostability of the Single-Electron-Transfer Complexes of Anabaena Ferredoxin-NADP(+) Reductase. Chemphyschem. 2015, 16, 3161-3169. https://doi.org/10.1002/cphc.201500534.
We agree and have added one sentence to the revised manuscript. Two suggested references were cited on page 4 of the revised manuscript.
4) “An increasing number of PSI structures of (…) showing a high resemblance between structures” (lines 61-63). Could the authors quantify the degree of resemblance between the aforemention PSI structures among the existing biology sources.
We are in the process of developing a computational method to quantify the resemblance of PS I protein 3D structures as well as pigment (e.g., chlorophyll) 3D structures. A structural analysis of nearly all PS I structures available in PDB will be a separate study submitted in the future.
5) METHODS. “To purify the PSI complexes (…) to a concentration of 1.5% (w/v)” (lines 250-252). Did the authors observe any aggregation effects during the purification process? In case affirmative, what strategies were followed in order to minimize this detrimental effect? Some insights should be provided in this regard.
We did not observe any aggregation effects during the purification procedure. After we isolate thylakoid membranes, we use the homogenizer to create a uniform and even mixture of thylakoid membranes before the solubilization step. The homogenization step is important. We added a sentence on homogenization in the Method section (page 12).
6) “The pump pulse (…) annihilation effects [37][REF]” (lines 333-335). Please, it lacks a reference citation at the end of this statement. The authors should fix this point.
We deleted “[REF]” from this sentence. The reference citation to this statement ([37]) was already present in the text.
7) Finally, a subsection related to the statistical analysis carried out by the authors should be added in Materials & Methods.
We agree. A short section on the statistical analysis was added in the revised manuscript on page 17.
8) RESULTS. This section unequivocally shows the data gathered by the authors in the different techniques used in this research. Did the authors carry out stopped flow measurements (or other analogous experiment) to discern the electron transfer reaction kinetics among the wild-type and the mutants and how the specific changes on the aminoacid residues could affect to the electron catalytic processes? Some information should be provided in this regard.
The focus of the manuscript was on the A0 cofactor that is directly involved in the early electron transfer steps induced by absorption of light. These steps occur in picosecond time range and are best characterized by optical femtosecond spectroscopy, which can both initiate charge transfer and monitor its progress with subpicosecond time resolution. Stop-flow technique cannot initiate charge transfer in PS I RC with sufficient temporal resolution and was thus not used. The overall effect of A0 mutations on light harvesting efficiency was characterized by the growth rate efficiency and oxygen evolution.
9) DISCUSSION. This section clearly states the most relevant outcomes found in this research. The authors should consider to add some potential future action lines to pursue this research.
We have added a paragraph highlighting potential future research direction related to this project on pages 30-31.

Reviewer 2 Report
Comments and Suggestions for Authors
The manuscript of Luo et al. ‘Impact of peripheral hydrogen bond on electronic properties of the primary acceptor chlorophyll in the reaction center of photosystem I’ was reviewed. In overall, the work is written well with sufficient descriptions of PSI structure and the electron transport futures.
In my opinion the manuscript can be accepted after some corrections, which I list below.
The authors write about the mutants ‘PsaA-M688H and PsaB-M668H’ (line 106), however these amino acids are not indicated in Fig 1. Can the authors add them, for example, in grey?
In the methods section I would recommend to add more detailed information about the approach with a bead-beater use (line 244). What beads were used (zircon, glass, or other)? Were there some moments with beads filling the chamber?
It would be appropriate to add relevant references about the method used for pigments quantification (line 271). In addition, I would like the authors to add reference to another work about long-lived P700+ (line 296) (https://doi.org/10.1134/S0006297920060073).
Is it sufficient a light intensity equal to 50 μE m-2 s-1 for O2 evolution measurement? (line 276). For example, in study with Chlamydomonas the light intensity of more than 2000 μmol m-2 s-1 is usually used (doi: 10.1007/s11120-020-00802-2). If this is correct for cyanobacteria, the authors should add a relevant reference here.
The authors indicate the same content of carotenoids in the studied strains (mutants) (fig S3b). However, in fig S4a the upper brown band related to carotenoids is obviously more intensive in the case of the double mutant. The authors should explain it.
The authors write about a slightly reduced PS I/PS II ratio based on the data from 77K, but the data from SDS-electrophoresis or Western blot would give more accurate information about PSI and PSII content. 77K indicates the size of antennas around PSII and PSI, which can be changed as result of state transitions and the authors can get incorrect data.
Author Response
Reviewer #2
The authors appreciate the insightful comments of reviewer #2 and hope that addressing them will improve the quality and clarity of the manuscript.
The authors write about the mutants ‘PsaA-M688H and PsaB-M668H’ (line 106), however these amino acids are not indicated in Fig 1. Can the authors add them, for example, in grey?
We have added relevant amino acids (notations PsaA-M684 and PsaB-M659 for Synechocystis sp. PCC 6803) to Fig. 1, as well as a clarifying sentence in the revised manuscript that explains different numeration of the amino acids between Synechocystis sp. PCC 6803 and Chlamydomonas reinhardtii (pages 5-6).
In the methods section I would recommend to add more detailed information about the approach with a bead-beater use (line 244). What beads were used (zircon, glass, or other)? Were there some moments with beads filling the chamber?
We have added information about the chambers and glass beads used to break the cells to the revised manuscript on page 12.
It would be appropriate to add relevant references about the method used for pigments quantification (line 271). In addition, I would like the authors to add reference to another work about long-lived P700+ (line 296) (https://doi.org/10.1134/S0006297920060073).
We have included the relevant references.
Is it sufficient a light intensity equal to 50 μE m-2 s-1 for O2 evolution measurement? (line 276). For example, in study with Chlamydomonas the light intensity of more than 2000 μmol m-2 s-1 is usually used (doi: 10.1007/s11120-020-00802-2). If this is correct for cyanobacteria, the authors should add a relevant reference here.
We checked the manual for the Chlorolab-2 oxygen electrode, and it states the light intensity is 900 μE m-2 s-1. We have corrected it on page 13. Thanks.
The authors indicate the same content of carotenoids in the studied strains (mutants) (fig S3b). However, in fig S4a the upper brown band related to carotenoids is obviously more intensive in the case of the double mutant. The authors should explain it.
The double mutant has slightly higher carotenoid content on average than the WT. However, it did not reach a significant difference (p = 0.65). We have added a sentence in the revised manuscript on page 20.
The authors write about a slightly reduced PS I/PS II ratio based on the data from 77K, but the data from SDS-electrophoresis or Western blot would give more accurate information about PSI and PSII content. 77K indicates the size of antennas around PSII and PSI, which can be changed as result of state transitions and the authors can get incorrect data.
We agree that a Western blotting experiment can better determine the PS I and PS II ratio. However, we do not have the capacity to perform Western blotting experiments. We have rephrased the sentences about the 77K fluorescence result (Supplementary Page 3). There are no obvious differences between the PS I trimer band and the band of PS I monomer and PS II mixture between the mutants and the WT.

Round 2
Reviewer 1 Report
Comments and Suggestions for Authors
The authors did a great deal of effort to cover all the suggestions raised by the Reviewers. For it, the scientific manuscript quality was greatly improved. Based on the significance of the most relevant outcomes found in this resesrch and the scope of IJMS, I warmly endorse this work for further publication in this journal.
Reviewer 2 Report
Comments and Suggestions for Authors
I am grateful to the authors for their responses and corrections made in the manuscript according to my comments. Thus, I think that the manuscript can be accepted for publication in the current view.